# Understanding and Improving Early Stopping for Learning with Noisy Labels

**Yingbin Bai**[1][*]    **Erkun Yang**[2][*]    **Bo Han**[3]    **Yanhua Yang**[2]
**Jiatong Li**[4]    **Yinian Mao**[4]    **Gang Niu**[5]    **Tongliang Liu**[1][†]

[1]TML Lab, University of Sydney; [2]Xidian University; [3]Hong Kong Baptist University;
[4]Meituan-Dianping Group; [5]RIKEN AIP

## Abstract

The memorization effect of deep neural network (DNN) plays a pivotal role in many state-of-the-art label-noise learning methods. To exploit this property, the early stopping trick, which stops the optimization at the early stage of training, is usually adopted. Current methods generally decide the early stopping point by considering a DNN as a whole. However, a DNN can be considered as a composition of a series of layers, and we find that the latter layers in a DNN are much more sensitive to label noise, while their former counterparts are quite robust. Therefore, selecting a stopping point for the whole network may make different DNN layers antagonistically affect each other, thus degrading the final performance. In this paper, we propose to separate a DNN into different parts and progressively train them to address this problem. Instead of the early stopping which trains a whole DNN all at once, we initially train former DNN layers by optimizing the DNN with a relatively large number of epochs. During training, we progressively train the latter DNN layers by using a smaller number of epochs with the preceding layers fixed to counteract the impact of noisy labels. We term the proposed method as progressive early stopping (PES). Despite its simplicity, compared with the traditional early stopping, PES can help to obtain more promising and stable results. Furthermore, by combining PES with existing approaches on noisy label training, we achieve state-of-the-art performance on image classification benchmarks. The code is made public at `https://github.com/tmllab/PES`.

## 1   Introduction

Deep networks have revolutionized a wide variety of tasks, such as image processing, speech recognition, and language modeling [7], However, this highly relies on the availability of large annotated data, which may not be feasible in practice. Instead, many large datasets with lower quality annotations are collected from online queries [5] or social-network tagging [18]. Such annotations inevitably contain mistakes or *label noise*. As deep networks have large model capacities, they can easily memorize and eventually overfit the noisy labels, leading to poor generalization performance [36]. Therefore, it is of great importance to develop a methodology that is robust to noisy annotations.

Existing methods on learning with noisy labels (LNL) can be mainly categorized into two groups: model-based and model-free algorithms. Methods in the first category mainly model noisy labels with the noise transition matrix [24, 34, 33, 30]. With perfectly estimated noise transition matrix, models trained with corrected losses can approximate to the models trained with clean labels. However,

---

[*]co-first author

[†]Correspondence to Tongliang Liu (tongliang.liu@sydney.edu.au)

35th Conference on Neural Information Processing Systems (NeurIPS 2021).

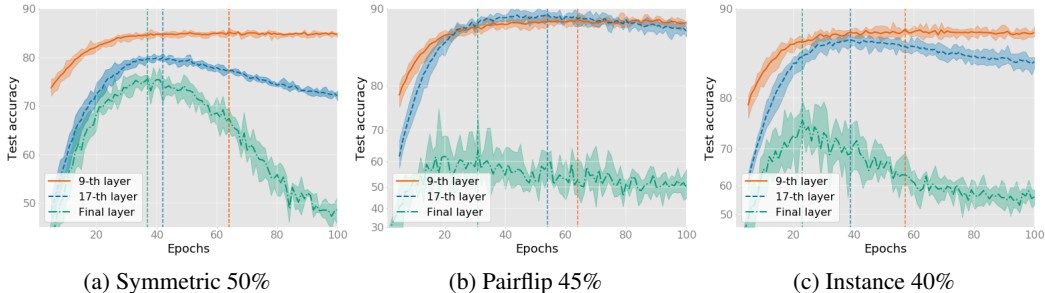

| (a) Symmetric 50% | (b) Pairflip 45% | (c) Instance 40% |

Figure 1: We train a ResNet-18 model on CIFAR-10 with three types of noisy labels and evaluate the impact of noisy labels on the representations from the 9-th layer, the 17-th layer, and the final layer. The X-axis is the number of epochs for the first block of the network. The curves present the mean of five runs and the best performances are indicated with dotted vertical lines.

current methods are usually fragile to estimate the noise transition matrix for heavy noisy data and are also hard to handle a large number of classes [9]. The second type explores the dynamic process of optimization policies, which relates to the memorization effect−deep neural networks tend to first memorize and fit majority (clean) patterns and then overfit minority (noisy) patterns [2]. Recently, based on this phenomenon, many methods [9, 26, 15, 16, 29] have been proposed and achieved promising performance.

To exploit the memorization effect, when the double descent phenomenon [3, 22, 11] cannot be guaranteed to occur, a core issue is to study when to stop the optimization of the network. While stopping the training for too few epochs can avoid overfitting to noisy labels, it can also make the network underfit to clean labels. Current methods [25, 23] usually adopt an early stopping strategy, which decides the stopping point by considering the network as a whole. However, since DNNs are usually optimized with stochastic gradient descent (SGD) with backpropagation, supervisory signals will gradually propagate through the whole network from latter layers (i.e., layers that are closer to output layers) to former layers (i.e., layers that are closer to input layers). Noting that the output layer is followed by the empirical risk in the optimization procedure. We hypothesize that noisy labels may have more severe impacts for the latter layers, which is different from current methods [9, 15] that usually stop the training of the whole network at once.

To empirically verify the above hypothesis, we analyze the impact of noisy labels on representations from different layers with different training epochs. To quantitatively measure the impact of noisy labels from intermediate layers, we first train the whole network on noisy data with different training epochs and fix the parameters for the selected layer and its previous layers. We then reinitialize and optimize the rest layers with clean data, and the final classification performance is adopted to evaluate the impact of noisy labels. For the final layer, we directly report the overall classification performance. As illustrated in Figure 1, we can see that latter layers always achieve the best performance at relatively smaller epoch numbers and then exhibit stronger performance drops with additional training epochs, which verifies the hypothesis that noisy data may have more severe impacts for latter layers. With this understanding, we can infer that the early stopping, which optimizes the network all at once, may fail to fully exploit the memorization effect and induce sub-optimal performance.

To address the above problem, we propose to optimize a DNN by considering it as a composition of several DNN parts and present a novel progressive early stopping (PES) method. Specifically, we initially train former DNN layers by optimizing them with a relatively large number of epochs. Then, to alleviate the impact of noisy labels for latter layers, we reinitialize and progressively train latter DNN layers by using smaller numbers of epochs with preceding DNN layers fixed. Since different layers are progressively trained with different early stopping epochs, we term the proposed method as progressive early stopping (PES). Despite its simplicity, compared with normal early stopping trick, PES can help to better exploit the memorization effect and obtain more promising and stable results. Moreover, since the model size and training epochs are gradually reduced during the optimization procedure, the training time of PES is only slightly greater than that of the normal early stopping. Finally, by combining PES with existing approaches on noisy label training tasks, we establish new state-of-the-art (SOTA) results on CIFAR-10 and CIFAR-100 with synthetic noise. We also achieve competitive results on one dataset with real-world noise: Clothing-1M [32].

The rest of the paper is organized as follows. In Section 2, we first introduce the proposed progressive early stopping and then present the details of the proposed algorithm by combining our method with existing approaches on noisy label training tasks. Section 3 shows the experimental results of our proposed method. Related works are briefly reviewed in Section 4. Finally, concluding remarks are given in Section 5.

## 2 Proposed Method

Let $D$ be the distribution of a pair of random variables $(\boldsymbol{X}, \boldsymbol{Y}) \in \mathcal{X} \times \{1, ...K\}$, where $\boldsymbol{X}$ indicates the variable of instances, $\boldsymbol{Y}$ is the variable of labels, $\mathcal{X}$ denotes the feature space, and $K$ is the number of classes. In many real-world problems, examples independently drawn from the distribution $D$ are unavailable. Before being observed, the clean labels are usually randomly corrupted into noisy labels. Let $\tilde{D}$ be the distribution of the noisy example $(\boldsymbol{X}, \tilde{\boldsymbol{Y}})$, where $\tilde{\boldsymbol{Y}}$ indicates the variable of noisy labels. For label-noise learning, we can only access a sample set $\{\boldsymbol{x}_i, \tilde{y}_i\}_{i=1}^n$ independently drawn from $\tilde{D}$. The aim is to learn a robust classifier from the noisy sample set that can classify test instances accurately.

In the following, we first elaborate on the proposed progressive early stopping (PES). Then, based on PES, we provide a learning algorithm that learns with confident examples and semi-supervised learning techniques.

### 2.1 Progressive Early Stopping

When trained with noisy labels, if clean labels are of majority within each noisy class, deep networks tend to first fit clean labels during an early learning stage before eventually memorizing the wrong labels, which can be explained by the memorization effect. Many current methods utilize this property to counteract the influence of noisy labels by stopping the optimization at an early learning phase. Specifically, a deep classifier can be obtained by optimizing the following objective function with a relatively small epoch number $T$:

$$\min_{\Theta} \frac{1}{n} \sum_{i=1}^{n} \mathcal{L}(f(\boldsymbol{x}_i; \Theta), \tilde{y}_i), \tag{1}$$

where $f(\cdot; \Theta)$ is a deep classifier with model parameters $\Theta$ and $\mathcal{L}$ is the cross-entropy loss. When trained with noisy data, early learning regularization (ELR) [16] reveals that, for the most commonly used cross-entropy loss, the gradient is well correlated with the correct direction at the early learning phase. Therefore, with a properly defined small epoch number $T$, the classifier can have higher accuracy than at initialization. While, if we continue to optimize the deep model after $T$ epochs, the classifier will be able to memorize more noise labels. Therefore, it is critical to select a proper epoch number $T$ to utilize the memorization effect and alleviate the influence of noisy labels.

Current methods typically select the epoch number $T$ by considering the network as a whole. However, as Figure 1 makes clear, the impact of noisy labels on different DNN layers are different, which implies that the traditional early stopping trick, which optimizes the whole network all at once, may make different DNN layers to be antagonistically affected by each other, thus degrading the final model performance. To this end, we propose to separate a DNN into different parts and progressively train layers in different parts with different training epochs. Specifically, assume that the whole network $f(\cdot; \Theta)$ can be constituted with $L$ DNN parts

$$\begin{aligned} \boldsymbol{z}_1 &= f_1(\boldsymbol{x}; \Theta_1), \\ \boldsymbol{z}_l &= f_l(\boldsymbol{z}_{l-1}; \Theta_l), \quad l = 2, \ldots, L \end{aligned} \tag{2}$$

where $f_l(\cdot; \Theta_l)$ is the $l$-th DNN part and $\boldsymbol{z}_l$ is the corresponding output. The output of the last part $\boldsymbol{z}_L$ is the prediction. The network $f(\cdot; \Theta)$ can also be represented as $f(\cdot; \Theta_1, ...\Theta_L)$. To counteract the impact of noisy labels, We initially optimize the parameter $\Theta_1$ for the first part by training the whole network for $T_1$ epochs with the following objective

$$\min_{\Theta_1...\Theta_k} \frac{1}{n} \sum_{i=1}^{n} \mathcal{L}(f(\boldsymbol{x}_i; \Theta_1, \ldots, \Theta_L), \tilde{y}_i). \tag{3}$$

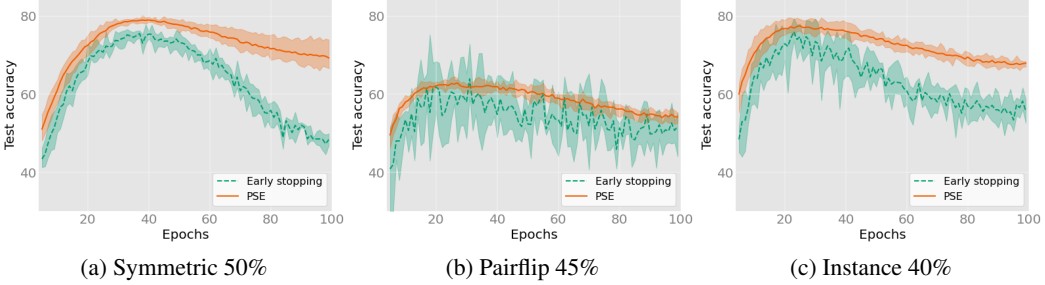

| (a) Symmetric 50% | (b) Pairflip 45% | (c) Instance 40% |

Figure 2: Performance of the traditional early stopping trick and the proposed PES on CIFAR-10 with different types of label noise. The lines present the mean of five runs.

Then, we keep the obtained parameter $\Theta_1^*$ fixed, reinitialize and progressively learn the $l$-th ($l = 2, \ldots, L$) DNN part with the parameters for preceding DNN parts fixed. The training procedure is conducted with $T_l$ epochs by optimizing the following objective

$$\min_{\Theta_l \ldots \Theta_k} \frac{1}{n} \sum_{i=1}^{n} \mathcal{L}(f(\boldsymbol{x}_i; \Theta_1^*, \ldots, \Theta_{l-1}^*, \Theta_l, \ldots, \Theta_L), \tilde{y}_i), \quad l = 2 \ldots L \tag{4}$$

We gradually optimize the $(l+1)$-th DNN part with the obtained parameter $\Theta_l^*$ fixed, the optimization is continued until all the parameters have been optimized. As elaborated above, latter DNN parts are more sensitive to noisy labels than their former counterparts. Therefore, for the above initializing optimization in Eq. (3) and the following $L - 1$ steps of optimization in Eq. (4), we gradually reduce the training epochs (i.e. $T_1 \geq T_2 \geq \cdots \geq T_L$) to better exploit the memorization effect. After optimization, we can obtain the final network as $f(\cdot, \Theta) = f(\cdot; \Theta_1^*, \ldots, \Theta_L^*)$. Since this model is obtained by progressively exploiting the early stopping strategies for different DNN parts, we term the proposed method as progressive early stopping (PES).

To explicitly verify the effectiveness of the proposed PES method, we conduct several pilot experiments, which compare the traditional early stopping and PES with label noise from different types and different levels. The results are illustrated in Figure 2, from which we can see that, compared with models trained with traditional early stopping, models trained with PES can achieve superior classification accuracy with smaller variations in all the cases. Current state-of-the-art methods [15] usually adopt models with the traditional early stopping as base models to distill confident examples and then utilize semi-supervised learning techniques by considering confident examples as labeled data and other noisy examples as unlabeled data to further improve the results. The final performance still heavily relies on the base model trained with noisy labels. By improving the performance of the base model, our method combined with semi-supervised learning techniques is able to establish new state-of-the-art results. In the following subsections, we will elaborate on how to utilize PES to distill confident examples and further combine it with semi-supervised learning techniques.

## 2.2 Learning with Confident Examples

Based on the deep network optimized with progressive early stopping, we can select confident examples to facilitate the model training. Here, confident examples refer to examples that have high probabilities with clean labels. In this paper, we treat examples whose predictions are consistent with given labels as confident examples. In addition, to make the results more robust, we generate two different augmentations for any given input and use the average prediction to decide its predicted label. Formally, we can obtain the confident example set $\mathcal{D}_l$ as

$$\mathcal{D}_l = \{(\boldsymbol{x}_i, \tilde{y}_i) | \tilde{y}_i = \hat{y}_i, i = 1, \ldots, n\},$$
$$\hat{y}_i = \underset{k \in \{1, \ldots, K\}}{\arg \max} \frac{1}{2} [f^k(\text{Augment}(\boldsymbol{x}_i); \Theta) + f^k(\text{Augment}(\boldsymbol{x}_i); \Theta)], \tag{5}$$

where $\text{Augment}(\cdot)$ indicates normal data augmentation operation including horizontal random flip and random crops, and $f^k(x; \Theta)$ is the predicted probability of $\boldsymbol{x}$ belonging to class $k$. Note that $\text{Augment}(\cdot)$ is a stochastic transformation, so the two terms in Eq (5) are not identical. The average

---
**Algorithm 1:** Progressive Early Stopping with Semi-Supervised Learning
---
**Input**: Neural network with trainable parameters $\Theta = \{\Theta_1, \ldots, \Theta_L\}$, Noisy training dataset $\{\boldsymbol{x}_i, \tilde{y}_i\}_{i=1}^n$, Number of training epochs for different part: $T_1, \ldots, T_L$, and training epochs $T_c$ for refining with confident examples.

**for** $i = 1, \ldots, T_1$ **do**
$\quad$ Optimize network parameter $\Theta$ with Eq. (3);
**for** $l = 2, \ldots, L$ **do**
$\quad$ Froze $\{\Theta_1, \ldots, \Theta_{l-1}\}$ and re-initialize $\{\Theta_l, \ldots, \Theta_L\}$;
$\quad$ **for** $i = 1, \ldots, T_l$ **do**
$\quad\quad$ Optimize network parameter $\{\Theta_l, \ldots, \Theta_L\}$ with Eq. (4);
Unfroze $\Theta$;
**for** $i = 1, \ldots, T_c$ **do**
$\quad$ Extract confident example set $\mathcal{D}_l$ and unlabeled set $\mathcal{D}_u$ with classifier $f(\cdot, \Theta)$ by Eq. (7);
$\quad$ Training the classifier $f(\cdot, \Theta)$ with MixMatch loss on $\mathcal{D}_l$ and $\mathcal{D}_u$;
Evaluate the obtained classifier $f(\cdot, \Theta)$.
---

prediction of augmented examples provides a more stable prediction and is found empirically to improve performance. After obtaining the confident example set, one can easily train a classifier by considering confident examples as clean data. However, since the number of confident examples for different classes can vary greatly, directly training the model with the obtained confident example set may introduce a severe class imbalance problem. To this end, we adopt a weighted classification loss

$$\mathcal{L}_c = \sum_{i=1}^{N} w_{y_i} \mathcal{L}_p(\tilde{y}_i, f(\boldsymbol{x}_i; \Theta)), \tag{6}$$

where $w_i$ is the corresponding class weight. Assuming that $\sigma_k = |\{(\boldsymbol{x}_i, \tilde{y}_i)|\tilde{y}_i = k, (\boldsymbol{x}_i, \tilde{y}_i) \in \mathcal{D}_l\}|$ denotes the cardinality of the confident example set belonging to the $k$-th class. Then, we can set $w_i = \sigma_i/(\sum_{j=1}^{K} \sigma_j)$ to indicate the corresponding class importance.

### 2.3 Combining with Semi-Supervised Learning

Training with only confident examples neglects the rest data and may suffer from insufficient training examples. To tackle this problem, we further resort to semi-supervised learning techniques by considering confident examples as labeled data and other noisy examples as unlabeled data. Specifically, the labeled data set and unlabeled data set can be obtained as

$$\begin{cases} \mathcal{D}_l = \{(\boldsymbol{x}_i, \tilde{y}_i)|\tilde{y}_i = \hat{y}_i, i = 1, \ldots n\} \\ \mathcal{D}_u = \{\boldsymbol{x}_i|\tilde{y}_i \neq \hat{y}_i, i = 1, \ldots n\} \end{cases}$$
$$\hat{y}_i = \underset{k \in \{1, \ldots K\}}{\arg\max} \frac{1}{2}[f^k(\text{Augment}(\boldsymbol{x}_i); \Theta) + f^k(\text{Augment}(\boldsymbol{x}_i); \Theta)], \tag{7}$$

where the labeled data set $\mathcal{D}_l$ is the same as that in Eq (6), and $\mathcal{D}_u$ is the rest unlabeled data set. Similar to [15], we adopt MixMatch [4] as the semi-supervised learning framework to train the final classification models. For more details about semi-supervised learning, we refer to [4]. The whole learning algorithm is summarized in Algorithm 1.

## 3 Experiments

### 3.1 Datasets and Implementation Details

**Datasets:** We evaluate our method on two synthetic datasets, CIFAR-10 and CIFAR-100 [12] with different levels of symmetric, pairflip, and instance-dependent label noise (abbreviated as instance label noise) and a real-world dataset Clothing-1M [32]. Both CIFAR-10 and CIFAR-100 contain 50k training images and 10k test images of size $32 \times 32$. Following previous works [9, 31, 16, 29], symmetric noise is generated by uniformly flipping labels for a percentage of the training dataset

Table 1: Preliminary analysis of the performance and the quality of extracted confident examples on CIFAR-10. The mean and standard deviation are computed over five runs.

| Metrics | Methods | Sym-20% | Sym-50% | Pair-45% | Inst-20% | Inst-40% |
|---|---|---|---|---|---|---|
| Test Accuracy | Early Stopping | 82.55±2.46 | 70.76±1.24 | 60.62±5.59 | 84.41±0.90 | 74.73±2.65 |
| | PES | **85.87±1.59** | **75.87±1.33** | **62.40±2.34** | **86.58±0.45** | **77.07±1.18** |
| Label Precision | Early Stopping | 98.81±0.15 | 94.65±0.19 | 72.53±5.26 | **98.70±0.43** | **90.77±1.87** |
| | PES | **98.96±0.09** | **95.46±0.14** | **72.99±2.27** | 98.52±0.19 | 90.63±0.92 |
| Label Recall | Early Stopping | 88.51±2.26 | 75.18±1.00 | 67.84±5.06 | 90.37±1.01 | 82.15±3.17 |
| | PES | **92.67±1.43** | **81.03±1.83** | **71.06±2.27** | **93.24±0.60** | **85.91±0.68** |

to all possible labels. Pairflip noise flips noisy labels into their adjacent class. And, instance noise is generated by image features. More details about the synthetic label noise are given in the *supplementary material*. For the flipping rate, it can include [9, 31] or ex-include [15, 16] true labels. We use the flipping rate including correct labels in Table 3 to compare with results in [15], and use without correct labels in the rest of the experiments. Clothing-1M [32] is a large-scale dataset with real-world noisy labels, whose images are clawed from the online shopping websites, and labels are generated based on surrounding texts. It contains 1 million training images, and 15k validation images, and 10k test images with clean labels.

**Baselines:** Semi-supervised learning may strongly boost the performance, we separately compare our method with approaches with or without semi-supervised learning. For the comparison with baselines with semi-supervised learning, we combine our proposed method with MixMatch used in [15] as indicated in Subsection 2.3. (1) Approaches without semi-supervised learning: Co-teaching [9], Forward [24], Joint Optim [25], T-revision [31], DMI [34], and CDR [29]. (2) Methods with semi-supervised learning: M-correction [1], DivideMix [15], and ELR+ [16]. We also adopt standard training with cross-entropy (CE) and MixUp [37] as baselines to show improvements.

**Network structure and optimization:** Our method is implemented by PyTorch v1.6. Baseline methods are implemented based on public codes with hyper-parameters set according to the original papers. For DivideMix and ELR+, we evaluate the test accuracy with the first network. To better demonstrate the robustness of our algorithm, we keep the hyper-parameters fixed for different types of label noise. More technique details are given in the *supplementary material*.

For experiments without semi-supervised learning, we follow [31], and use ResNet-18 [10] for CIFAR-10 and ResNet-34 for CIFAR-100. We split networks into three parts, the layers above block 4 as part 1, block 4 of ResNet as part 2, and the final layer as part 3. $T_1$ is defined as 25 for CIFAR-10 and 30 for CIFAR-100, $T_2$ as 7, and $T_3$ as 5. The network is trained for 200 epochs and SGD with 0.9 momentum is used. The initial learning rate is set to 0.1 and decayed with a factor of 10 at the 100th and 150th epoch respectively, and a weight decay is set to $10^{-4}$. For $T_2$ and $T_3$, we employ an Adam optimizer with a learning rate of $10^{-4}$.

For experiments with semi-supervised learning, we follow the setting of [15] with PreAct Resnet-18. We set the final layer as part 2, the rest as part 1. $T_1$ is defined as 20 for CIFAR-10 and 35 for CIFAR-100, and $T_2$ as 5. The network is trained for 300 epochs. For optimization, we use a single cycle of *cosine annealing* [19], and the learning rate begins from $2 \times 10^{-2}$ and ends at $2 \times 10^{-4}$, with a weight decay of $5 \times 10^{-4}$. An Adam optimizer is adopted with a learning rate of $10^{-4}$ for $T_2$. For hyper-parameters from MixMatch, we set them according to the original paper [4].

For Clothing-1M [32], we follow the previous work [25], and employ a ResNet-50 [10] pre-trained on ImageNet [13]. We set the final layer as part 2, the rest as part 1. $T_1$ and $T_2$ are defined as 20 and 7 respectively. The network is trained with CE loss for 50 epochs and SGD is used with 0.9 momentum and a weight decay of $10^{-3}$. The learning rate is $5 \times 10^{-3}$ and decayed by a factor of 10 at the 20th and 30th epoch respectively. We employ an Adam optimizer with a learning rate of $5 \times 10^{-6}$ for $T_2$.

### 3.2 Preliminary Experiments

In Figure 2, we can observe that with the PES trick, the performance of classifiers is generally improved compared with that the traditional early stopping trick. In this section, we further carefully analyze the quality of extracted labels by examining them from three aspects, i.e., test accuracy,

Table 2: Comparison with state-of-the-art methods without semi-supervised learning on CIFAR-10 and CIFAR-100. The mean and standard deviation computed over five runs are presented.

| Dataset | Method | Symmetric | | Pairflip | Instance | |
|---|---|---|---|---|---|---|
| | | 20% | 50% | 45% | 20% | 40% |
| CIFAR10 | CE | 84.00±0.66 | 75.51±1.24 | 63.34±6.03 | 85.10±0.68 | 77.00±2.17 |
| | Co-teaching | 87.16±0.11 | 72.80±0.45 | 70.11±1.16 | 86.54±0.11 | 80.98±0.39 |
| | Forward | 85.63±0.52 | 77.92±0.66 | 60.15±1.97 | 85.29±0.38 | 74.72±3.24 |
| | Joint Optim | 89.70±0.11 | 85.00±0.17 | 82.63±1.38 | 89.69±0.42 | 82.62±0.57 |
| | T-revision | 89.63±0.13 | 83.40±0.65 | 77.06±6.47 | 90.46±0.13 | 85.37±3.36 |
| | DMI | 88.18±0.36 | 78.28±0.48 | 57.60±14.56 | 89.14±0.36 | 84.78±1.97 |
| | CDR | 89.72±0.38 | 82.64±0.89 | 73.67±0.54 | 90.41±0.34 | 83.07±1.33 |
| | Ours | **92.38±0.40** | **87.45±0.35** | **88.43±1.08** | **92.69±0.44** | **89.73±0.51** |
| CIFAR100 | CE | 51.43±0.58 | 37.69±3.45 | 34.10±2.04 | 52.19±1.42 | 42.26±1.29 |
| | Co-teaching | 59.28±0.47 | 41.37±0.08 | 33.22±0.48 | 57.24±0.69 | 45.69±0.99 |
| | Forward | 57.75±0.37 | 44.66±1.01 | 27.88±0.80 | 58.76±0.66 | 44.50±0.72 |
| | Joint Optim | 64.55±0.38 | 50.22±0.41 | 42.61±0.61 | 65.15±0.31 | 55.57±0.41 |
| | T-revision | 65.40±1.07 | 50.24±1.45 | 41.10±1.95 | 60.71±0.73 | 51.54±0.91 |
| | DMI | 58.73±0.70 | 44.25±1.14 | 26.90±0.45 | 58.05±0.20 | 47.36±0.68 |
| | CDR | 66.52±0.24 | 55.30±0.96 | 43.87±1.35 | 67.33±0.67 | 55.94±0.56 |
| | Ours | **68.89±0.45** | **58.90±2.72** | **57.18±1.44** | **70.49±0.79** | **65.68±1.41** |

Table 3: Comparison with state-of-the-art methods with semi-supervised learning on CIFAR-10 and CIFAR-100 with symmetric label noise from different levels. Results with * are token from [15]. The mean and standard deviation are computed over three runs.

| Dataset | CIFAR-10 | | | CIFAR-100 | | |
|---|---|---|---|---|---|---|
| Methods / Noise | Sym-20% | Sym-50% | Sym-80% | Sym-20% | Sym-50% | Sym-80% |
| CE | 86.5±0.6 | 80.6±0.2 | 63.7±0.8 | 57.9±0.4 | 47.3±0.2 | 22.3±1.2 |
| MixUp | 93.2±0.3 | 88.2±0.3 | 73.3±0.3 | 69.5±0.2 | 57.1±0.6 | 34.1±0.6 |
| M-correction* | 94.0 | 92.0 | 86.8 | 73.9 | 66.1 | 48.2 |
| DivideMix* | 95.2 | 94.2 | 93.0 | 75.2 | 72.8 | 58.3 |
| DivideMix | 95.6±0.1 | 94.6±0.1 | 92.9±0.3 | 75.3±0.1 | 72.7±0.6 | 56.4±0.3 |
| ELR+ | 94.9±0.2 | 93.6±0.1 | 90.4±0.2 | 75.5±0.2 | 71.0±0.2 | 50.4±0.8 |
| Ours (Semi) | **95.9±0.1** | **95.1±0.2** | **93.1±0.2** | **77.4±0.3** | **74.3±0.6** | **61.6±0.6** |

label precision, and label recall. Here, label precision indicates the ratio of the number of extracted confident examples with correct labels in the total confident example set, and label recall represents the ratio of the number of confident examples with correct labels among the total correctly labeled examples. Specifically, we train a neural network on CIFAR-10 with different kinds and levels of label noise for 25 epochs respectively and report the performance for each case before and after the proposed PES is applied.

Results in Table 1 clearly show that, compared with the traditional early stopping, PES can help to obtain higher accuracies, precisions, and recalls for most cases. For instance-dependent label noise, PES can achieve higher recall values with comparable label precision values. Note that models with high recall values can help to collect more confident examples, which is critical for learning with confident examples and semi-supervised learning. Therefore, by enhancing the performance of the initial model, PES can help to improve the final classification performance in all cases, which is also verified by the experiments in Section 3.3.

## 3.3 Classification Accuracy Evaluation

**Synthetic datasets.** We first verify the effectiveness of our proposed method without semi-supervised learning techniques on two synthetic datasets: CIFAR-10 and CIFAR-100. For both of these two datasets, we leave 10% of data with noisy labels as noisy validation set. Results are presented in Table 2, which shows that our proposed method can consistently outperform all other baselines across various settings by a large margin.

Table 4: Comparison with state-of-the-art methods with semi-supervised learning on CIFAR-10 and CIFAR-100 with instance-dependent and pairflip label noise from different levels. The mean and standard deviation are computed over three runs.

| Dataset | CIFAR-10 | | | CIFAR-100 | | |
|---|---|---|---|---|---|---|
| Methods / Noise | Inst-20% | Inst-40% | Pair-45% | Inst-20% | Inst-40% | Pair-45% |
| CE | 87.5±0.5 | 78.9±0.7 | 74.9±1.7 | 56.8±0.4 | 48.2±0.5 | 38.5±0.6 |
| MixUp | 93.3±0.2 | 87.6±0.5 | 82.4±1.0 | 67.1±0.1 | 55.0±0.1 | 44.2±0.5 |
| DivideMix | 95.5±0.1 | 94.5±0.2 | 85.6±1.7 | 75.2±0.2 | 70.9±0.1 | 48.2±1.0 |
| ELR+ | 94.9±0.1 | 94.3±0.2 | 86.1±1.2 | 75.8±0.1 | 74.3±0.3 | 65.3±1.3 |
| Ours (Semi) | **95.9±0.1** | **95.3±0.1** | **94.5±0.3** | **77.6±0.3** | **76.1±0.4** | **73.6±1.7** |

Table 5: Compassion with state-of-the-art methods on Clothing-1M. Results of baseline methods are taken from the original papers. ours represent the results obtained by PES with a single network and ours* indicate the results obtained by PES with an ensemble model.

| CE | Forward | Joint-Optim | DMI | T-revision | DivideMix* | ELR+* | Ours | Ours* |
|---|---|---|---|---|---|---|---|---|
| 69.21 | 69.84 | 72.16 | 72.46 | 74.18 | 74.76 | 74.81 | **74.64** | **74.99** |

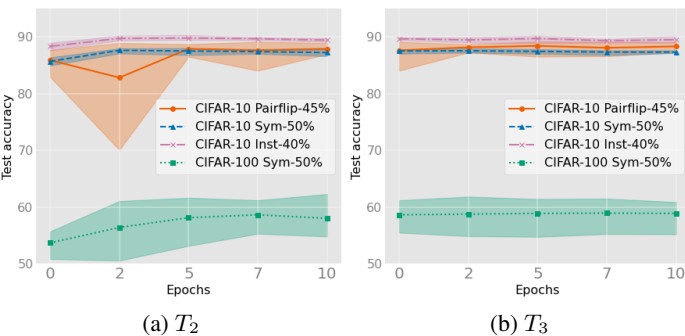

(a) $T_2$        (b) $T_3$

Figure 3: Sensitivity analysis for different training iteration numbers: $T_2$ and $T_3$.

Table 3 and Table 4 present the mean accuracy and standard deviation for our method and all baselines on CIFAR-10 and CIFAR-100, respectively. From the results, we can get that the proposed method can outperform all baselines in all cases. For pairflip label noise, the advantages of our proposed method become more apparent, and it significantly outperforms state-of-the-art methods by over 8% on both CIFAR-10 and CIFAR-100. These empirical results support our proposal that PES can improve the quality of selected confident examples, which helps improve performance and reduce the variance of the final classifier.

**Real-world dataset.** We evaluate the performance of the proposed method on a real-world dataset with Clothing-1M [32] and select methods such as CE, Forward, Joint-Optim, DMI, and T-revision, which use a single network, and also methods such as DivideMix and ELR+, which adopt an ensemble model with two different networks, as baselines. We also report the results for the proposed PES with a single network as *ours* and the results for PES, which ensembles two networks, as *ours\**. The overall results are reported in Table 5, from which we can observe that the proposed PES with a single network can outperform all baselines using a single network. And with an ensemble model, which contains two different networks, our method can outperform all the adopted baselines. These results clearly demonstrate that, by improving the performance of the initial classification network, our method is more flexible to handle such real-world noise problems.

### 3.4 Sensitivity Analysis

In this section, we investigate the hyper-parameter sensitivity for the training iteration number $T_2$ and $T_3$, respectively. We firstly analyze the training epoch number for the second DNN part by varying $T_2$ from the range of $[0, 10]$. The results are illustrated in Figure 3a, from which can find that, with

Table 6: Training time comparison for baselines on CIFAR-10 with 50% Symmetric label noise.

| CE | Co-teaching | CDR | T-revision | ELR+ | DivideMix | Ours | Ours (Semi) |
|------|-------------|------|------------|------|-----------|------|-------------|
| 0.9h | 1.5h | 3.0h | 3.5h | 2.2h | 5.5h | 1.0h | 3.1h |

the increasing of $T_2$, the performance of PES first increase and then decrease in all the cases except for 45% Pairflip noise on the CIFAR-10 dataset. While the model achieves the best performance with $T_2$ as 7 for all types of noisy labels. Then we fix $T_2$ as 7, and analyze the impact of the third DNN part by varying $T_3$ from the range of $[0, 10]$. The results are shown in Figure 3b. Although the performance variance for different $T_3$ is smaller than that for $T_2$, we can still observe that the best performance can be obtained when $T_3$ is set as 5. More importantly, from these two figures, we can get that both $T_2$ and $T_3$ are robust to the different types of noisy labels.

## 3.5 Training Time Comparison

In this section, we compare the training time of our method and other state-of-the-art baselines. All the experiments are conducted on a server with a single Nvidia V100 GPU. The training times for all the methods are reported in Table 6, from which we can get that our algorithm with cross-entropy loss achieves the fastest speed across all baselines, only about 1 hour. Our method combining with MixMatch [4] is also fast, only a little more than half of the training time of DivideMix. The time of ELR+ [16] shows superior, but ELR+ trains the network with fewer epochs, with 200 epochs compared with ours for 300 epochs.

## 4 Related work

Learning with noisy data has been well studied [17, 6, 21, 27, 20]. Current works can be mainly categorized into two groups: model-based and model-free methods. In this section, we briefly review some closely related works.

The first type models the relationship between clean labels and noisy labels by estimating the noise transition matrix and build a loss function to correct the loss [24, 30, 35, 28]. [24] first combines algorithms for estimating the noise rates and loss correction techniques together and introduces two alternative procedures for loss correction. It also proves that both of the two procedures enjoy formal robustness guarantees *w.r.t.* the clean data distribution. DMI [34] proposes an information-theoretic loss function, which utilizes Shannon's mutual information and is robustness to different kinds of label noise. T-revision [31] estimates the noise transition matrix without anchor points by adding a fine-tuned slack variables. Although these methods have made certain progress, they are usually fragile to estimate the noise transition matrix for heavy noisy data and are also hard to handle a large number of classes. Therefore, in this paper, we mainly focus on the model-free methods.

The second strand mainly counteracts noisy labels by exploiting the memorization effect that deep networks tend to first memorize and fit majority (clean) patterns and then overfit minority (noisy) patterns [2]. To exploit this property, Co-teaching [9] employs two networks with different initialization and uses *small loss* to select confident examples. M-correction [1] uses two Gaussian Mixture Models to identify confident examples, instead of using networks themselves. DivideMix [15] extends Co-teaching [9] and employs two Beta Mixture Model to select confident examples. MixMatch [4] is then adopted to leverage unconfident examples with a semi-supervised learning framework. All the above methods exploit the memorization effect by considering the adopted network as a whole. Recently, [14] shows that networks training with noisy labels can produce good representations, if the structure of networks suits the targeted tasks. Our method further explains that noisy labels have different impacts for different layers in a DNN. And latter layers will receive earlier and more severe impact than their former counterparts. Therefore, by considering a DNN as a composition of several layers and training different layers with different epochs, our method is able to better exploit the memorization effect and achieve superior performance.

## 5 Conclusion

In this work, we provide a progressive early stopping (PES) method to better exploit the memorization effect of deep neural networks (DNN) for noisy-label learning. We first find that the impact of noisy

labels for former layers in a DNN is much less and later than that for latter DNN layers, and then build upon this insight to propose the PES method, which separates a DNN into different parts and progressively train each part to counteract the different impacts of noisy labels for different DNN layers. To show that PES can boost the performance of state-of-the-art methods, we conduct extensive experiments across multiple synthetic and real-world noisy datasets and demonstrate that the proposed PES can help to obtain substantial performance improvements compared to current state-of-the-art baselines. The main limitation of our method lies in that, by splitting a DNN into different parts, PES introduces several additional hyper-parameters that need to be tuned carefully. In the future, we will extend the work in the following aspects. First, we will study other mechanisms that distinguishing desired and undesired memorization rather than early stopping, e.g., the gradient ascent trick [8]. Second, we are interested in combining PES with interesting ideas from semi-supervised learning and unsupervised learning.

## Acknowledgments and Disclosure of Funding

YB was partially supported by Agriculture Consultant and Smart Management. BH was supported by the RGC Early Career Scheme No. 22200720, NSFC Young Scientists Fund No. 62006202 and HKBU CSD Departmental Incentive Grant. YY was partially supported by Key Research and Development Program of Shaanxi (ProgramNo. 2021ZDLGY01-03). GN was supported by JST AIP Acceleration Research Grant Number JPMJCR20U3, Japan. TL was partially supported by Australian Research Council Projects DE-190101473 and IC-190100031.

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
