# Understanding and Improving Early Stopping for Learning with Noisy Labels Supplementary

**Yingbin Bai**[1][*]    **Erkun Yang**[2][*]    **Bo Han**[3]    **Yanhua Yang**[2]
**Jiatong Li**[4]    **Yinian Mao**[4]    **Gang Niu**[5]    **Tongliang Liu**[1][†]

[1]TML Lab, University of Sydney; [2]Xidian University; [3]Hong Kong Baptist University;
[4]Meituan-Dianping Group; [5]RIKEN AIP

## A    Training details

In this section, we first provide details about the adopted three kinds of noisy labels. Then, we elaborate on the data preprocessing and the hyperparameter settings in our experiments.

### A.1    Definition of noise

According to different correlations between noisy labels and clean labels, there are three kinds of widely used label noise, namely symmetric class-dependent label noise, pairflip class-dependent label noise, and instance-dependent label noise [9, 3, 10]. In the following, we first introduce one basic concept: transition matrix [9], and then provide the details for all the three kinds of label noise, respectively.

**Transition matrix:** The *transition matrix* $T(\boldsymbol{x})$ is used to explicitly model the generation process of label noise, where $T_{ij}(\boldsymbol{x}) = \Pr(\bar{Y} = j | Y = i, X = \boldsymbol{x})$ is the flip rate between the true label and noisy label on given data $\boldsymbol{x}$. $X$ is the variable of instances, $Y$ is the variable of clean labels, and $\bar{Y}$ is the variable of noisy labels. $T_{ij}(\boldsymbol{x})$ is the $ij$-th entry of the transition matrix $T(\boldsymbol{x})$, which denotes the probability of the instance $\boldsymbol{x}$ with clean label $i$ being observed with a noisy label $j$.

**Symmetric class-dependent label noise:** Symmetric class-dependent label noise is generated with symmetric class-dependent noise transition matrices. We set the flip rate $\alpha$. Random flipping labels may change to true labels, so the flip rate may include or exclude true labels. For the flip rate excluding true labels, the diagonal entries of symmetric transition matrix are $1 - \alpha$ and the off-diagonal entries are $\alpha/(c-1)$. For the flip rate including true labels, the diagonal entries of symmetric transition matrix are $1 - (\alpha \times (c-1)/c)$ and the off-diagonal entries are $\alpha/c$.

**Pairflip class-dependent label noise:** Pairflip noise is a simulation of fine-grained classification with noisy labels, where annotators may make mistakes only within very similar classes[13, 8]. The label noise is generated with pairflip class-dependent noise transition matrices, which is defined as follow. Let flip rate is $\alpha$. The diagonal entries of a pairflip transition matrix are $1 - \alpha$ and the entities for their adjacent classes, which the examples in a given class may be wrongly classified to, are $\alpha$.

**Instance-dependent label noise:** We generate the instance-dependent label noise according to Algorithm 1. More details about this algorithm can be found in [10].

### A.2    Data preprocessing and experimental settings

**Data preprocessing:** For experiments on CIFAR-10/100 [5] without semi-supervised learning, we use simple data augmentation techniques including random crop and horizontal flip. For experiments

---

[*]co-first author

[†]Correspondence to Tongliang Liu (tongliang.liu@sydney.edu.au)

35th Conference on Neural Information Processing Systems (NeurIPS 2021).

---
**Algorithm 1:** Instance-dependent Label Noise Generation
---
**Input**: Clean samples $\{(\boldsymbol{x}_i, y_i)\}_{i=1}^n$; Noise rate $\tau$.
1: Sample instance flip rates $q \in \mathbb{R}^n$ from the truncated normal distribution $\tau\mathcal{N}(\tau, 0.1^2, [0, 1])$;
2: Independently sample $w_1, w_2, \ldots, w_c$ from the standard normal distribution $\mathcal{N}(0, 1^2)$;
3: For $i = 1, 2, \ldots, n$ do
4:    $p = \boldsymbol{x}_i \times w_{y_i}$;                          // generate instance-dependent flip rates
5:    $p_{y_i} = -\infty$;         // control the diagonal entry of the instance-dependent transition matrix
6:    $p = q_i \times softmax(p)$; // make the sum of the off-diagonal entries of the $y_i$-th row to be $q_i$
7:    $p_{y_i} = 1 - q_i$;                                          // set the diagonal entry to be $1 - q_i$
8:    Randomly choose a label from the label space according to possibilities $p$ as noisy label $\bar{y}_i$;
9: End for.
---
**Output**: Noisy samples $\{(\boldsymbol{x}_i, \bar{y}_i)\}_{i=1}^n$
---

on CIFAR-10/100 with semi-supervised learning, except random cropping and horizontal flip, MixUp [14] is also employed, which is a critical component of MixMatch [1]. For Clothing-1M [12], we first resize images to $256 \times 256$, and then random crop to $224 \times 224$, following a random horizontal flip.

**Hyper-parameters of PES**: We adopt an Adam optimizer for $T_2$ and $T_3$ for accelerating the model training and reducing the parameter turning, and $T_2$ and $T_3$ are chosen from $\{2, 5, 7\}$. Note that the number of total training epochs includes $T_1$, but excludes $T_2$ and $T_3$. To make PES work in large datasets, we regard training $100,000$ examples as an epoch in Clothing1M experiments.

**Hyper-parameters of semi-supervised learning**: We keep all the hyper-parameters fixed for different levels of noise, and only adjust $\lambda_u$ for different noisy settings, since the ratio of confident examples (labeled data) and unconfident examples (unlabeled data) can vary greatly for different noisy settings. Specifically, we set $K = 2, T = 0.5$, and $\lambda_u$ is chosen from $\{5, 15, 25, 50, 75, 100\}$. $\alpha$ begins with $4$, and changes to $0.75$ after 150th epoch. More details of hyper-parameters can be found in Table 1 and Table 2.

Table 1: Training hyper-parameters for CIFAR-10/100 and Clothing-1M

|  | CIFAR-10 | | CIFAR-100 | | Clothing-1M |
|---|---|---|---|---|---|
| architecture | ResNet-18 | PreAct ResNet-18 | ResNet-34 | PreAct ResNet-18 | Pretrained Resnet-50 |
| loss function | CE | MixMatch loss | CE | MixMatch loss | CE |
| learning rate (lr) | 0.1 | 0.02 | 0.1 | 0.02 | $5 \times 10^{-3}$ |
| lr decay | 100th & 150th | Cosine Annealing | 100th & 150th | Cosine Annealing | 20th & 30th |
| weight decay | $10^{-4}$ | $5 \times 10^{-4}$ | $10^{-4}$ | $5 \times 10^{-4}$ | $10^{-3}$ |
| batch size | 128 | 128 | 128 | 128 | 64 |
| training examples | 45,000 | 50,000 | 45,000 | 50,000 | 1,000,000 |
| training epochs | 200 | 300 | 200 | 300 | 50 |
| PES lr | $10^{-4}$ | $10^{-4}$ | $10^{-4}$ | $10^{-4}$ | $5 \times 10^{-6}$ |
| $T_1$ | 25 | 20 | 30 | 35 | 20 |
| $T_2$ | 7 | 5 | 7 | 5 | 7 |
| $T_3$ | 5 | - | 5 | - | - |

Table 2: Semi-supervised loss weight $\lambda_u$ for CIFAR-10/100

| Datasets / Noise | Sym-20% | Sym-50% | Sym-80% | Pairflip-45% | Inst-20% | Inst-40% |
|---|---|---|---|---|---|---|
| CIFAR-10 | 5 | 15 | 25 | 5 | 5 | 15 |
| CIFAR-100 | 50 | 75 | 100 | 50 | 50 | 50 |

# B    Additional experiments

In this section, we provide more experimental results on CIFAR-100 and Fashion-MNIST to further verify the hypothesize that noisy labels may have more severe impacts on the latter layers. We also provide additional comparisons with baselines, which exploit ensemble networks.

In the first experiment, we adopt a dataset with more classes: CIFAR-100 and a deeper network: ResNet-34 [4]. In addition, we adopt Fashion-MNIST [11], including 60,000 training images with 28x28 size and LeNet [2], which consists of two convolutional layers and three full-connected layers

with ReLU activation. The learning procedure for CIFAR-100 and Fashion-MNIST is the same as that for CIFAR-10 in the paper. Specifically, we first train the whole network on noisy data with different training epochs. For the final layer, we directly report the overall classification performance. For other selected layers, we frozen the parameters for the selected layer and previous layers, and then reinitialize and optimize the rest layers with clean data, and the final classification performance is adopted to evaluate the impact of noisy labels. We do not use image augmentation techniques for Fashion-MNIST dataset.

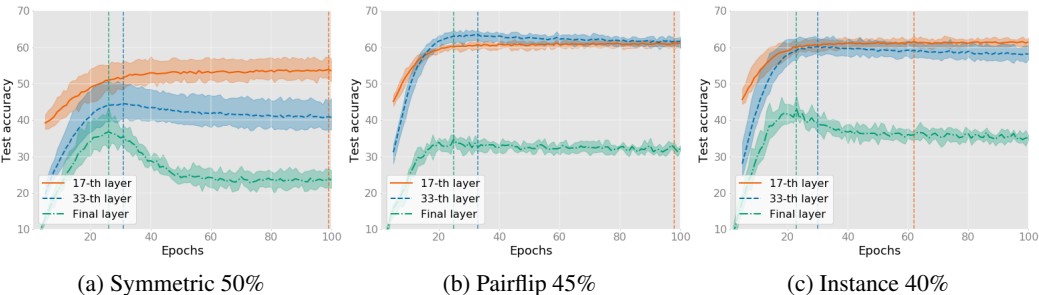

|  (a) Symmetric 50% | (b) Pairflip 45% | (c) Instance 40% |

Figure 1: We adopt ResNet-34 as the model on CIFAR-100 and evaluate the impact of noisy labels on the representations from the 17-th layer, the 33-th layer, and the final layer. The curves present the mean of five runs and the best performances highlight with dotted vertical lines.

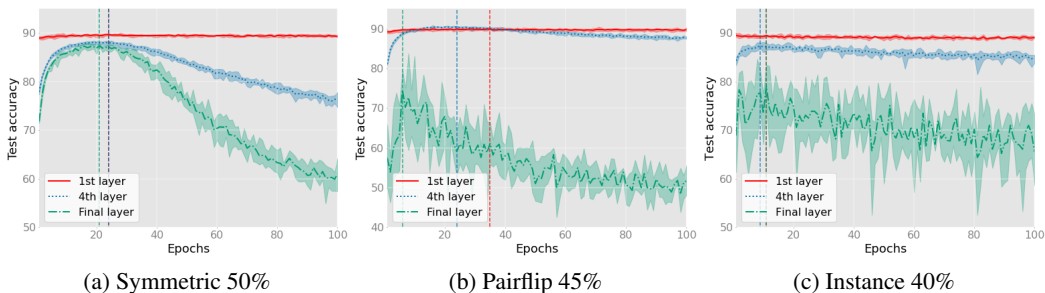

|  (a) Symmetric 50% | (b) Pairflip 45% | (c) Instance 40% |

Figure 2: We adopt LeNet as the model on Fashion-MNIST and evaluate the impact of noisy labels on the representations from the 1-st layer, the 4-th layer, and the final layer. The curves present the mean of five runs and the best performances highlight with dotted vertical lines. Note that vertical lines are merged together for the 1-st layer and 4-th layer on Symmetric 50%, and vertical lines of the 1-st layer and the final layer are merged together on Instance 40%.

Figure 1 and Figure 2 demonstrate the impacts of noisy labels on different layers on CIFAR-100 and Fashion-MNIST, respectively. From Figure 1, we can see that the drop of the green line (the final layer) is the largest, the blue line (the 33-th layer) has a gradual decline, and the orange line (the 17-th layer) is relatively stable during the training process. These observations are similar to those for CIFAR-10. The performance of 17-th layer in ResNet-34 is affected by noisy labels later and less than that of the 9-th layer in ResNet-18. It is because there are more layers after the 17-th layer in ResNet-34 than the 9-th layer. Similar trends are observable in Figure 2. The first layer is nearly unaffected by noisy labels, and the performance of the final layer has a larger decline compared with the 4-th layer. The learning speeds of different layers are unapparent in LeNet, since there are only three hidden layers in LeNet, and the gradient of losses transfers much easier compared with deeper networks. Another reason may be the simplicity of patterns in Fashion-MNIST without image augmentation techniques, which leads the convolutional layers to learn fast.

In the paper, we compare our results with baselines evaluated with a single network. In this section, we compare our method with state-of-the-art methods with ensemble two networks taken from the original papers [6, 7]. We also adopt cross-entropy and MixUp [14] with a single network as baselines. From Table 3, we can observe our results with a single network are comparable to results of baselines with ensemble two networks. Specifically, on CIFAR-100, our method outperforms state-of-the-art methods across all settings.

Table 3: Comparison with state-of-the-art methods using ensemble two networks and semi-supervised learning on CIFAR-10 and CIFAR-100 with symmetric label noise from different levels. Baseline results are taken from [6] and [7]. The highest results are reported for all the methods.

| Dataset | CIFAR-10 | | | CIFAR-100 | | |
|---|---|---|---|---|---|---|
| Methods / Noise | Sym-20% | Sym-50% | Sym-80% | Sym-20% | Sym-50% | Sym-80% |
| CE | 87.2 | 80.9 | 65.8 | 58.1 | 47.5 | 23.6 |
| MixUp | 93.5 | 88.4 | 73.6 | 69.7 | 57.9 | 34.69 |
| DivideMix* | **96.1** | 94.6 | 93.2 | 77.3 | 74.6 | 60.2 |
| ELR+ | 95.8 | 94.8 | **93.3** | 77.6 | 73.6 | 60.8 |
| Ours (Semi) | **96.1** | **95.3** | **93.3** | **77.7** | **74.9** | **62.3** |