# OpenReview forum: "Understanding and Improving Early Stopping for Learning with Noisy Labels"
_NeurIPS.cc/2021/Conference — NeurIPS 2021 Poster_

### Official Review · Reviewer_9vjN · 2021-07-13

**Rating:** 6
**Confidence:** 4

**Summary:**

The paper introduces a novel training scheme (progressive early stopping, PES) aimed at improving the performance of convolutional networks for image classification in presence of noisy labels. The strategy is motivated by the observation that the layers close to the output tend to overfit noisy targets more than early layers. PES thus proposes to differentiate the number of training epochs for different groups of layers, training early groups for more epochs than final ones. In detail PES divides the networks in N blocks and trains for T_1 epochs the whole network. It then freezes the learned weights of the first block and  reinitializes  and trains the layers of the remaining N-1 blocks for T_2 epochs. Crucially, T_2 < T_1. The same policy is then progressively repeated for all the remaining N-2 blocks with T_n <T_n-1 for n in  {1, …, N}.
PES is also combined with a semi-supervised scheme (MixMatch) to reach SOTA performances on CIFAR10 and CIFAR100 datasets.


**Limitations And Societal Impact:**

The limitations are rather briefly but rather clearly discussed at the end of the paper.

**Main Review:**

Quality:
1)The training policy described in lines 192-196 is quite unclear. Here the number of epochs is set to 200 and the learning rate, starting from 0.1, is decreased by a factor 10 at epochs 100 and 150. Then T1 is set to 20/35, T2 to 7, T3 to 5 and the learning rate is set to 10**-4 at T2 and T3. What happens after T1+T2+T3 epochs? The value of the learning rate after the first T1 epochs is 0.1 or 10**-4?

2) What is the motivation for introducing two different optimizers, i.e. SGD with a step learning rate scheduler and Adam in the same training run (see lines 192-209)?

3) I have a concern regarding a comparison between the results reported in Table2 and the sensitivity analysis shown in Fig.3. If I understood correctly, PES with T2=0 and T3=0 should be equivalent to the standard cross entropy training since the whole network is trained for T1 epoch with noisy labels. From Table2  we can see that the accuracy gap between CE and PES for, say, cifar100 trained with symmetric noise 50% is between 16/21%. From Figure 3 instead we see that the accuracy gap between the optimal T2/T3 and T2=T3=0 is well below 10%.  A similar concern regards the other perturbations shown in the sensitivity analysis.
Why is this happening?

4) As correctly recognized by the author(s) the approach requires fixing several metaparameters, in particular the number of epochs for each sub-block optimization. But why is this necessary? The common practice in early stop is finishing the optimization close to the maximum of the  accuracy estimated on a small dataset not used for the optimization (and different from the validation set). Why cannot this procedure be used in PES?

5) The code used to train the network should be publicly shared.

Clarity:

The paper could be improved in the following parts:

When the authors describe Fig. 1 they should say on which dataset they train the network used to produce the profiles, (only looking at table 3 of section 3.2 it is possible to guess that the dataset is Cifar10), and for how many epochs they trained the first block of the network. Moreover the meaning of ‘clean data’ line 53 is not very clear. Indeed if one compares the accuracy of a classifier trained only on a noisy  dataset with that obtained with the same classifier on noisy and noiseless data it seems quite natural that in the second case the accuracy improves;

Line 172: it is not clear what ‘adiacent class’ means;

There are a few typos and grammar errors.


**Time Spent Reviewing:**

4

---

> ### Author Response · Authors · 2021-08-07
> **Response to Reviewer 9vjN**
>
> **Q1, The code used to train the network should be publicly shared.**
>
> A1, We agree that algorithm implementation is very important. We have attached the source code in supplementary materials, with which, all the results in our paper can be re-implemented. Moreover, we will release all the source code once the paper is accepted.
> &nbsp;
> &nbsp;
>  **  * ** * ** * **
> **Q2, The training policy described in lines 192-196 is quite unclear ... What happens after T1+T2+T3 epochs? The value of the learning rate after the first T1 epochs is 0.1 or 10\*\*-4?**
>
> A2, 200 is the total training epochs including the classifier training with PES and the subsequent learning with confident examples. The first T1+T2+T3 epochs are for the classifier training with PES. After that, we will continue the learning with confident examples. The whole learning procedure can also be found in Algorithm 1 in the original paper.
>
> After T1, the learning rate of PES is set to 10\*\*−4, while for learning with confident examples part, the learning rate is first set to 0.1 and gradually decreased by a factor of 10 at the 100th and 150th epochs, respectively. We apologize for the confusion and would like to clarify them in the revised version.
> &nbsp;
> &nbsp;
>  **  * ** * ** * **
> **Q3, What is the motivation for introducing two different optimizers, i.e. SGD with a step learning rate scheduler and Adam in the same training run (see lines 192-209)?**
>
> A3, As is verified in many empirical studies in other related tasks, with proper tuning, SGD with momentum can usually achieve better performance than Adam. Therefore, we adopt SGD with momentum in most procedures.
>
> While, as is demonstrated in the paper, the last two blocks of a DNN are more sensitive to noisy labels, which motivates us to alternatively adopt Adam optimizer to accelerate the model training and reduce the parameter tuning, thus alleviating the critical overfitting problem. We have also conducted experiments, which verify that adopting Adam for the last two blocks can help to achieve superior performance. We will elaborate on this in the revised version.
> &nbsp;
> &nbsp;
>  **  * ** * ** * **
> **Q4, I have a concern regarding a comparison between the results reported in Table 2 and the sensitivity analysis shown in Fig.3. ... Why is this happening?**
>
> A4, PES with T2=0 and T3=0 is equal to traditional early stopping. The main difference between PES with T2=0 and T3=0 in Figure 3 and CE in Table 2 is that PES employs learning with confident examples method, while CE uses the standard training process. Therefore, the accuracy gaps between Table 2 and Figure 3 are different. We will clarify this issue in the revised version.
> &nbsp;
> &nbsp;
>  **  * ** * ** * **
> **Q5, As correctly recognized by the author(s) the approach requires fixing several metaparameters, in particular the number of epochs for each sub-block optimization. But why is this necessary? ... Why cannot this procedure be used in PES?**
>
> A5, Thanks for the valuable comments. We can adopt the procedure in PES. However, in this paper, we target at the challenging learning with noisy labels task. Using a noisy dataset to estimate the maximum of the accuracy will introduce large result variance, which may degrade the final performance. Actually, we have conducted experiments with stopping at the maximum of the accuracy estimated on a small noisy dataset. The obtained performance is inferior to the strategy in our method.
>
> Considering all of the above, we adopt the strategy in our method, which introduces several additional parameters to control the training of the sub-blocks. We will introduce this in more detail in the revised version.
> &nbsp;
> &nbsp;
>  **  * ** * ** * **
> **Q6, When the authors describe Fig. 1 they should say on which dataset they train the network used to produce the profiles ... and for how many epochs they trained the first block of the network.**
>
> A6, We will describe the used dataset (Cifar10) in the revised version. The number of epochs for the first block of the network is the X-axis in Figure 1.
> &nbsp;
> &nbsp;
>  **  * ** * ** * **
> **Q7, The meaning of ‘clean data’ line 53 is not very clear. Indeed if one compares the accuracy of a classifier trained only on a noisy dataset with that obtained with the same classifier on noisy and noiseless data it seems quite natural that in the second case the accuracy improves;**
>
> A7, Our target in Figure 1 is not to quantitatively compare between three lines but to evaluate the impact of noisy labels on DNN blocks trained with different epochs. For example, to analyze the impact of noisy labels on block 1, we first train the whole network with noisy data, and then fix the parameters of block 1 and re-train the parameters of other layers with clean data. If the final performance is better, we can get that the impact of noisy labels on block 1 is less. We will explain this issue in more detail in the revised version.
> &nbsp;
> &nbsp;
>  **  * ** * ** * **
> **Q8, Line 172: it is not clear what ‘adiacent class’ means;**
>
> A8, Pairflip noise is a simulation of fine-grained classification with noisy labels, where annotators may make mistakes only within very similar classes [1,2]. And the adjacent class indicates the class that the examples in a given class may be wrongly classified to.
> &nbsp;
> &nbsp;
> [1] Yueming Lyu and Ivor W. Tsang. Curriculum loss: Robust learning and generalization against label corruption. In 8th International Conference on Learning Representations, ICLR 2020
>
> [2] Xingrui Yu, Bo Han, Jiangchao Yao, Gang Niu, Ivor W. Tsang, and Masashi Sugiyama.How does disagreement help generalization against label corruption? In Proceedings of the 36th International Conference on Machine Learning, ICML 2019,
>
>  **  * ** * ** * **
> &nbsp;
> **Thanks again for all the insightful comments. In the above, we have tried our best to provide point-to-point responses to all the comments and would also like to answer any further questions.**
> &nbsp;
> &nbsp;

---

> > ### Comment · Reviewer_9vjN · 2021-08-26
> > **change in my score**
> >
> > The authors replied satisfactory to most of the points I raised. I am a bit concerned by the reply to  Q5, which actually points to a possible weakness of the approach: the only manner of making it work properly seems to be fine-tuning the metaparameters in a manner that, I am afraid, will depend very much on the prediction task. I have however decided to raise my score to 6.

---

> ### Author Response · Authors · 2021-08-23
> **Response to Reviewer 9vjN**
>
> Dear Reviewer 9vjN,
>
> We have tried our best to address all the concerns and provided explanations to all questions. If there are still unclear parts to you, please kindly let us know. We are very glad to further discuss them.
>
> Best,
> Authors

---

### Official Review · Reviewer_sdsb · 2021-07-14

**Rating:** 7
**Confidence:** 3

**Summary:**

The paper tackles the problem of classification under a noisy data setting. The authors make the observation that noisy labels mainly affect the latter layers of deep neural networks, and propose a method to address this, termed progressive early stopping (PES). A network is trained by freezing the parameters layers of layers 1,2,...,L (numbered from input to output) at predefined epochs E1, E2...EL. After this step, the authors use the network thus trained to identify “confident” training examples (those for which the predicted label matches the true label). Then the paper proposes two ways of training a final model: (1) in a fully supervised fashion, using only the confident examples (while accounting for class imbalance), or (2) semi-supervised, where the examples that are not confident are considered unlabeled. The approach is tested on noisy versions of CIFAR10 and CIFAR100 and the Clothing-1M dataset, and compared with competing supervised and semi-supervised methods.


**Ethical Concerns:**

Nothing I can think of.


**Limitations And Societal Impact:**

Yes, the major limitations that I could think of are the running time, number of hyper parameters and sensitivity to hyper-parameters, which the authors have addressed. The other potential issue, I have mentioned in question 3 above.

I don’t think societal impact is discussed, but I don’t think this is a concern for this type of method and evaluation setting.


**Main Review:**

Overall, I found this paper interesting and well written. It proposes some interesting ideas and provides thorough comparisons with competing methods. The authors also addressed several issues that could be considered weaknesses such as training run time. On the negative side, I did not find the proposed “progressive early stopping” (PES) strategy very original, as it bears a close resemblance to greedy layer-wise training [1,2], and the authors do not discuss this similarity. However, even if this part is not novel, the authors do use it in what I think is a novel way to handle noisy labels. One aspect that I am a bit unclear about is whether the PES part is necessary for the second part of the paper starting from section 2.2.  -- what if you skip that part and compute the confidences in (5) using a normally trained baseline and then apply the proposed semi-supervised approach? Aside from this, the experimental analysis looks pretty good.

I will discuss this in more detail below.

**Originality:**  As mentioned above, the PES component is very similar in spirit to [1] and [2] (there are probably other papers too, but these are pretty popular), and while the methods may not be identical, without a comparison in the paper is hard to understand what is new. Using the predicted noisy labels as unlabeled data in semi-supervised learning is also not new, and it was proposed in DivideMix (cited as [15] by the authors). So in fact the combination of PES and the idea in DivideMix is what in fact makes this method work so well.

[1] Bengio, Yoshua, et al. "Greedy layer-wise training of deep networks." Advances in neural information processing systems. 2007.

[2] Belilovsky, Eugene et al. “Greedy Layerwise Learning Can Scale to ImageNet.” ICML 2019

**Quality**: I really appreciated the experimental analysis in this paper, which considers many baselines, does a sensitivity analysis on the hyperparameters and reports run times. The paper contains a lot of details about hyperparameter configurations, although it’s not clear to me how these were set, especially the number of epochs T1, T2… which also differ per dataset. Could the authors please clarify this?

**Significance**: The tackled problem is definitely of considerable practical importance, and the authors report significant improvements over existing methods. I am not though familiar with the most recent approaches for noisy labels to assess if the competing methods are state-of-the-art; perhaps the other reviewers can address this.
The method does have some limitations that hinder its use in practice, such as longer training time especially for the semi-supervised version, and the fact that there are multiple stages to it. However, this may be a trade-off that some users may be willing to pay for the improved performance.

**Some clarification questions:**
1. Line 161: how can we make sure we have enough labeled samples? For a bad classifier with low accuracy, many of the samples will be put in the unlabeled set, to the point where you may have too little data to train a good classifier using semi-supervised learning. What is a minimum accuracy necessary for the baseline classifier (the one used to compute the \hat y) in order for this method to have enough labeled samples to perform better than the baseline itself?
2. How were the hyper-parameters T1, T2… set in the various experiments? What about the rest of the training hyperparameters, such as learning rate, regularization etc.?
3. Eq. (5) Do the authors use only 2 random augmentations? Are two enough to get a reliable estimate? How sensitive is it to how much you alter the original image?
4. For methods “CE” in Tables 2 and 3? Is it standard training with cross-entropy on the noisy dataset? If so, how do you make this semi-supervised.
5. In case this is not covered by the previous question, what happens if you compute \hat y_i with early stopping in eq. (5), and then apply your semi-supervised approach on this? Is this reported in the table?


**Time Spent Reviewing:**

3h

---

> ### Author Response · Authors · 2021-08-10
> **Response to Reviewer sdsb**
>
> **Q1, The relationship between existing greedy layer-wise training methods and PES. & Paper originality.**
>
> A1, Although existing greedy layer-wise training methods seem similar to the proposed PES, we would like to clarify that they are indeed two different kinds of methods.
>
> Greedy layer-wise training first trains a small network and then gradually adds additional layers to form a deeper network. This technique is originally proposed to solve the optimization difficulties of Deep Belief Networks (DBN). While, PES always considers different DNN blocks as parts of the whole network and is proposed to better exploit the memorization effect to counteract the influence of noisy labels.
>
> Moreover, we are the first study that investigates the different impacts of noisy labels on different DNN layers, which reveals that deeper layers are more sensitive to noisy labels than shallower layers. We think our work can provide a new understanding of the impact of noisy labels and also brings new insights to the community of learning with noisy labels.
>
>  **  * ** * ** * **
>
> **Q2, One aspect that I am a bit unclear about is whether the PES part is necessary for the second part of the paper starting from section 2.2. – what if you skip that part and compute the confidences in (5) using a normally trained baseline and then apply the proposed semi-supervised approach? 5. In case this is not covered by the previous question, ... Is this reported in the table?**
>
> A2, Thanks, we can skip PES and directly combine early stopping with the adopted semi-supervised learning technique.
>
> Actually, early stopping and the adopted semi-supervised learning are basic components of many existing methods. For example, DivideMix[4] adopts early stopping and similar semi-supervised learning techniques as its main components. Besides, it also employs a Gaussian Mixture Model to better collect confident examples to improve the final performance. Therefore, although we do not directly report the performance of the model purely trained with early stopping and semi-supervised learning, the performance of DivideMix in Table 3 can be used as a comparison.
>
>
>  **  * ** * ** * **
>
> **Q3, Line 161: how can we make sure we have enough labeled samples? For a bad classifier with low accuracy, many of the samples will be put in the unlabeled set, to the point where you may have too little data to train a good classifier using semi-supervised learning.**
>
> A3, For learning with noisy labels, we always assume that clean data occupies the majority of a class. Otherwise, we cannot learn an effective classifier without any additional information. Recent works [1][2] reveal that, if clean data is the majority of a class, deep classifiers will firstly fit to clean data before they overfit to noisy data, which is also called the memorization effect of DNNs and can help to effectively collect confident examples.
>
> Actually, the number of the collected labeled examples can vary greatly due to different data geometrical structures and different noisy types. However, various experiments have shown that methods based on the memorization effect can collect a certain number of labeled examples, which are usually useful for semi-supervised learning. **Therefore, although it is hard to theoretically define how many labeled examples will be enough, this may not be a major problem in practice.**
>
>
>
>  **  * ** * ** * **
>
> **Q4, What is a minimum accuracy necessary for the baseline classifier (the one used to compute the ˆ y) in order for this method to have enough labeled samples to perform better than the baseline itself?**
>
> A4, Similar to the above question, it is hard to provide an exact minimum accuracy for the baseline classifier, since this value may vary from task to task. However, with the assumption that clean data occupies the majority of a class, the memorization effect can help to train a relatively good classifier[2]. The adopted semi-supervised learning technique has also been verified in various settings[3][4]. Therefore, in practice, the main concern is alternative how to obtain a better classifier to further improve the performance.
>
>
>
>  **  * ** * ** * **
>
> **Q5, How were the hyper-parameters T1, T2. . . set in the various experiments? What about the rest of the training hyperparameters, such as learning rate, regularization etc.?**
>
> A5, The basic principle to set the parameters of T1 to TL is to gradually reduce the training epochs for the latter DNN blocks. In our paper, we manually select the parameters of T1 to TL. Those parameters for CIFAR10 and CIFAR100 are set to be the same, which shows promising performance. Moreover, if a clean validation set is available, more advancing techniques, such as meta learning, can also be employed to select these parameters.
>
> Other hyper-parameters, including learning rate, weight decay, and momentum, are introduced in Section 3.1. Some other details e.g. data preprocessing and detailed settings for the semi-supervised learning are provided in Supplementary Material.
>
>
>  **  * ** * ** * **
>
>
> **Q6, Eq. (5) Do the authors use only 2 random augmentations? Are two enough to get a reliable estimate? How sensitive is it to how much you alter the original image?**
>
> A6, In this paper, we only use two augmentations and found that using two augmentations can already achieve promising results. Similar settings have also been used in [3][5].
>
> The sensitivity to data augmentation is an interesting problem and has been studied in self-supervised learning[6]. However, it is beyond the scope of our paper.
>
>
>  **  * ** * ** * **
>
>
>  **Q7, For methods “CE” in Tables 2 and 3? Is it standard training with cross entropy on the noisy dataset? If so, how do you make this semi-supervised.**
>
>  A7, Correct. In Tables 2 and 3, CE presents standard training with cross-entropy on the noisy dataset. The main difference between CE in Tables 2 and 3 lies in the adopted model architecture. Since CE is not a state-of-the-art method, we do not apply semi-supervised learning with it. We apologize that the title of Table 3 may make some confusion and would better clarify this in the revised version.
>
>
>  **  * ** * ** * **
>
>
> [1] Bo Han, Quanming Yao, Xingrui Yu, Gang Niu, Miao Xu, Weihua Hu, Ivor Tsang,and Masashi Sugiyama.  Co-teaching: Robust training of deep neural networks with extremely noisy labels. In NeurIPS, pages 8527–8537, 2018.
>
> [2] Sheng Liu, Jonathan Niles-Weed, Narges Razavian, and Carlos Fernandez-Granda.Early-learning regularization prevents memorization of noisy labels. In NeurIPS, 2020.
>
> [3] David Berthelot, Nicholas Carlini, Ian J. Goodfellow, Nicolas Papernot, Avital Oliver, and Colin Raffel.  Mixmatch: A holistic approach to semi-supervised learning.  In NeurIPS, pages 5050–5060, 2019.
>
> [4] Junnan Li, Richard Socher, and Steven C.H. Hoi. Dividemix: Learning with noisy labels as semi-supervised learning. In ICLR,2020.
>
> [5] Samuli Laine and Timo Aila. Temporal ensembling for semi-supervised learning. In ICLR, 2017.
>
> [6] Xiao Wang and Guo-Jun Qi. Contrastive learning with stronger augmentations, 2021.URLhttps://openreview.net/forum?id=KJSC_AsN14.

---

> > ### Comment · Reviewer_sdsb · 2021-08-27
> > **Thank you for your comments and clarifications!**
> >
> > I thank the authors for the detailed comments and clarifications! I have carefully read the authors' response, and my concerns were addressed. Thus, I will keep my accept score.

---

### Official Review · Reviewer_E13N · 2021-07-15

**Rating:** 7
**Confidence:** 5

**Summary:**

This paper presents an interesting new method namely progressively early stopping to better exploit the memorization effect of DNN for learning with noisy labels. It first analyzes different effects of noisy labels to different DNN layers and finds that latter layers are much more sensitive to noisy labels. Based on this understanding, the authors then propose a progressively early stopping method, which progressively stops the training of different layers to counteract the influence of noisy labels. By combining some existing semi-supervised learning techniques, the proposed method achieves SoTA performance on several popular image classification benchmarks.

**Ethical Concerns:**

No concern

**Limitations And Societal Impact:**

No concern was found. This paper should have positive social impact.

**Main Review:**

Pros:
-	Strong and clear motivation. The authors first analyze the influence of noisy labels on different DNN layers during different training iterations, which shows that latter layers are much more sensitive to noisy labels. Then, with this understanding, the authors propose to train different layers with different early stopping iterations to better exploit the memorization effect. The authors are the first to analyze the memorization effect for different DNN layers.
-	Solid technical contribution. The authors first split a DNN into different parts and optimize each part with different training iterations. Since latter layers are verified to be more sensitive to label noise, by applying smaller training iterations, the obtained model can be less overfitted to noisy labels. To exploit this phenomenon, the authors propose a new method dubbed progressively early stopping, which is also demonstrated to be easily combined with other semi-supervised learning techniques.
-	State of the art experimental performance. Their simple approach shows promising results and yields new SoTA performance on both synthetic and real-world datasets.
-	Convincing ablation experiments. The authors adopt a base CNN model and compare traditional early stopping with the proposed progressively early stopping. The results confirm that progressively early stopping is superior to the alternative.
-	Presentation is clear and easy to follow.

Cons:
-	The authors claim that different DNN layers have different sensitivities to noisy labels. Fig.1 demonstrates this point by splitting the DNN into three parts, while in the experiments, the authors alternatively split the adopted DNN into two parts. According to the paper, how the network is split seems important to the proposed method but is not introduced in detail.
-	In table 1, according to label precision, PES cannot surpass traditional early stopping, which seems not consistent with the claim of the method. And this issue is not explained in detail. Therefore, a better explanation is expected.
-	In Table 2, The proposed method seems always performs even better than other baselines when the noise is heavier under all types of noises. Authors should elaborate on this phenomenon to better understand the PES method.

**Time Spent Reviewing:**

3

---

> ### Author Response · Authors · 2021-08-10
> **Response to Reviewer E13N**
>
> **Q1, The authors claim that different DNN layers have different sensitivities to noisy labels. Fig.1 demonstrates this point by splitting the DNN into three parts, while in the experiments, the authors alternatively split the adopted DNN into two parts. According to the paper, how the network is split seems important to the proposed method but is not introduced in detail.**
>
> A1, Figure 1 aims to analyze the impact of noisy labels on different DNN layers and adopts CIFAR10 for the illustration. As explained in the paper, the number of DNN blocks is not fixed and can be adaptively set according to the dataset and DNN architectures. We would like to better explain this in the revised version.
>
>
>  **  * ** * ** * **
>
> **Q2, In table 1, according to label precision, PES cannot surpass traditional early stopping, which seems not consistent with the claim of the method. And this issue is not explained in detail. Therefore, a better explanation is expected.**
>
> A2, As presented in Table 1, PES can consistently outperform early stopping for symmetric and pairflip noises with all of the three metrics. For instance-dependent noise, PES achieves similar label precisions with early stopping, while clearly outperforms early stopping with the other two metrics. Therefore, we can obtain that PES can outperform early stopping for all the settings.
>
>
>  **  * ** * ** * **
>
> **Q3, In Table 2, The proposed method seems always performs even better than other baselines when the noise is heavier under all types of noises. Authors should elaborate on this phenomenon to better understand the PES method.**
>
> A3, Datasets with higher noise rates may induce more severe overfitting problems, thus degrading the final performance. Therefore, when the data noise becomes heavier, a clear performance drop is observed for most baselines. While, by progressively stopping the training of different DNN layers, our method can better exploit the memorization effect to counteract the overfitting problem. Hence, our method can outperform other baselines with even larger gaps under heavier label noises. This phenomenon clearly demonstrated the superiority of our method.
>
>
>  **  * ** * ** * **

---

> > ### Comment · Reviewer_E13N · 2021-08-30
> > **I'll keep my original score**
> >
> > I have carefully read the response. The authors have addressed my concerns well. Thus, I am glad to keep my score for acceptance. Thanks.

---

### Official Review · Reviewer_NpAD · 2021-07-16

**Rating:** 6
**Confidence:** 4

**Summary:**

In this manuscript, the authors discussed the sensitivity of deeper and shallower layers in a DNN and improved the conventional early stop of the whole model to a layer-wise progressive early stop (PES) for label-noise learning.

**Limitations And Societal Impact:**

Yes.

**Main Review:**

The idea is interesting. The authors improved the conventional early stop method in the label noise learning task. However, I’m confused with the task setting. Furthermore, the proposed method is too naive and the authors did not provide key part of the proposed method (see the third point in Limitations And Societal Impact). The related work was adequately cited. The manuscript is well-written. The experimental results seem good on two small-scale datasets, but the authors did not evaluate their method on ImageNet, which is a common setting.

1. In figure 1, the shallower layers are much better than the final layer with much higher accuracies in each case. How to understand this statement? Meanwhile, compared with figure 2, the test accuracies of the 9th layer are much higher than those of the PSE. Should we say that we can directly use the outputs of the 9th layer of the ResNet18, rather than the final outputs? This determines the significance of the label noise learning task.
2. As in figure 1, the shallower layers converge later than the deeper layers, which means the shallower ones should train in more epochs. However, the authors first froze shallower layers in eq (4) and algorithm 1. They are the opposite. How to explain this?
3. The authors did not discuss how to select T_1 to T_L in section 2.1, which is truly important. This makes the proposed method naive and hard to follow.



**Time Spent Reviewing:**

2

---

> ### Author Response · Authors · 2021-08-09
> **Response to Reviewer NpAD**
>
> **Q1, I’m confused with the task setting.**
>
> A1, Methods on learning with noisy labels (LNL) usually accept noisily labeled data as inputs, and aim to output models that can generalize well on clean test data.
>
> Current methods on LNL can be typically categorized into two groups: (1) model-based methods, which model noisy labels with the noisy transition matrix; (2) model-free methods, which mainly exploit the memorization effect. Our method belongs to the second type.
>
>
>
>  **  * ** * ** * **
>
> **Q2, The experimental results seem good on two small-scale datasets, but the authors did not evaluate their method on ImageNet, which is a common setting.**
>
> A2, To evaluate the model performance, two types of noisy data are usually employed. The first is hand-crafted noisy datasets and the other is real-world noisy datasets.
>
> (1) For the first type, CIFAR-10 and CIFAR-100 are the two most widely used datasets, since it is relatively easy to manufacture different kinds of noises with them.  In this paper, we adopt  symmetric, pairflip, and instance-dependent label noises from various noisy levels to evaluate our method; (2) For the second type, the most widely recognized dataset is a real-world large-scale noisy dataset: Clothing1M, which contains over 1 million images.
>
> In this paper, we follow the most common settings [1][2][3][4][5], and adopt CIFAR-10, CIFAR-100, and Clothing1M to demonstrate our method.
>
>
> [1] Daiki Tanaka, Daiki Ikami, Toshihiko Yamasaki, and Kiyoharu Aizawa. Joint optimiza-tion framework for learning with noisy labels. In CVPR, pages 5552–5560, 2018.
>
> [2] Yilun Xu, Peng Cao, Yuqing Kong, and Yizhou Wang.  L_dmi: A novel information-theoretic loss function for training deep nets robust to label noise.  In NeurIPS, 2019
>
> [3] Junnan Li, Richard Socher, and Steven C.H. Hoi. Dividemix: Learning with noisy labelsas semi-supervised learning. In ICML, 2020
>
> [4] Yu Yao, Tongliang Liu, Bo Han, Mingming Gong, Jiankang Deng, Gang Niu, andMasashi Sugiyama. Dual T: reducing estimation error for transition matrix in label-noiselearning. In NeurIPS, 2020.
>
> [5] Sheng  Liu,  Jonathan  Niles-Weed,  Narges  Razavian,  and  Carlos  Fernandez-Granda.Early-learning regularization prevents memorization of noisy labels. In NeurIPS, 2020.
>
>
>  **  * ** * ** * **
>
> **Q3, In figure 1, the shallower layers are much better than the final layer with much higher accuracies in each case. How to understand this statement?**
>
> A3, Thanks for your valuable comments. We would like to clarify that, in Figure 1, the results indicated as "9-th layer", "17-th layer", and "final layer" are not the performance of the representations directly generated  from these layers. For example, for the line of "9-th layer" in Figure 1, we first train the first 9 layers with different epochs and then train the rest layers with clean data. The final performance is illustrated as the line for "9-th layer". Results for other lines are obtained similarly.  These results are illustrated to investigate the impact of noisy labels on different DNN layers.
>
> As explained above, the results for shallower layers are obtained by first training shallower layers with noisy data and then training the other layers with clean data. While, since there are no following layers after the final layer, the results for "final layer" are obtained by training all the layers with noisy data. Therefore, it's reasonable that the results for the shallower layers are much better than those for the final layer.
>
>
>  **  * ** * ** * **
>
> **Q4, Meanwhile, compared with figure 2, the test accuracies of the 9th layer are much higher than those of the PSE. Should we say that we can directly use the outputs of the 9th layer of the ResNet18, rather than the final outputs? This determines the significance of the label noise learning task.**
>
> A4, Also, since in Figure 1, the results of "9-th layer" are obtained by first training the first 9 layers with noisy data and then training the rest layers with clean data. Its results are much better than those of PES trained with only noisy data. However, these results cannot be used for the learning with noisy label tasks, since clean data are used when they are obtained.
>
>
>  **  * ** * ** * **
>
> **Q5, As in figure 1, the shallower layers converge later than the deeper layers, which means the shallower ones should train in more epochs. However, the authors first froze shallower layers in eq (4) and algorithm 1. They are the opposite. How to explain this?**
>
> A5, We first train the whole model with larger epochs and froze the shallower layers. As indicated in Figure 1, deeper layers converge earlier than shallow layers. Therefore, deeper layers may already over-fit to noisy data. To address this problem, we  re-initialize the deeper layers and train them with a relatively small number of epochs. So, the effective training epochs for deeper layers are smaller than those for shallower layers.
>
>
>  **  * ** * ** * **
>
> **Q6, The proposed method is too naive.**
>
> A6, We would like to emphasize our contributions in the following three aspects:
>
> (1) We are the first study that investigates the different impacts of noisy labels on different DNN layers, which reveals that deeper layers are more sensitive to noisy labels than shallower layers. This study
> provides new understanding of the impact of noisy labels and also brings new insights to better exploit the memorization effect;
> (2) Based on the first study, we develop a novel progressively early stopping method, which can effectively alleviate the overfitting problem by gradually stopping the training of different DNN layers;
> (3) Extensive experiments on both synthetic and real-world noisy datasets are conducted, which show that our proposed method can significantly outperform current state-of-the-art methods.
>
> We do believe that our method has non-trivial contributions and will attract certain interests in the learning with noisy labels community.
>
>
>  **  * ** * ** * **
>
> **Q7, The authors did not provide key part of the proposed method ... The authors did not discuss how to select $T_1$ to $T_L$ in section 2.1, which is truly important. This makes the proposed method naive and hard to follow.**
>
> A7, Thank you for your valuable comments. The basic principle to set the parameters of  $T_1$ to $T_L$ is to gradually reduce the training epochs for latter DNN blocks.
>
> In our paper, we manually select the parameters of $T_1$ to $T_L$. Those parameters for CIFAR10 and CIFAR100 are the same, which show promising performance. Moreover, if a clean validation set is available, more advancing techniques, such as meta learning, can also be employed to further improve the method.

---

> > ### Comment · Reviewer_NpAD · 2021-08-20
> > **Thanks for your nice response.**
> >
> > Hi, I have carefully read all comments and the authors' feedback. I appreciate the authors’ detailed explanation and clarification, and the rebuttal has solved most of my concerns. I rechecked the paper again, the proposed PES makes sense, and the empirical results seem good. Given the current condition, I am happy to raise my score and lean on the positive side. But I strongly suggest that the authors should cover this to strengthen the clarity and contributions of this paper in the final version.

---

### Author Response · Authors · 2021-08-10
**Response to all reviewers**

Dear reviewers:

Thanks a lot for your efforts in reviewing this paper. We have tried our best to address all the concerns. Do you have any further questions? We will explain them in detail.

Best,
Authors

---

### Decision · Program_Chairs · 2021-09-27

**Decision:**

Accept (Poster)

**Comment:**

The paper proposes a variant of early stopping for learning with noisy labels. The method starts with the hypothesis that early layers are less prone to be adversely affected by label noise than later layers. Authors empirically validate this hypothesis and then propose a progressive layerwise training method for tackling label noise. This is then used to identify noisy examples which are treated as unlabeled data and existing semi-supervised methods are used to get further empirical gains. Reviewers have expressed some concerns on the heuristic components in the method (use of Adam for last layers vs SGD for earlier layers), several metaparameters (the number of epochs for each subblock), and the increased training time of the method due to progressive/multi-stage training, but they have found the motivation behind the paper convincing and have been overall positive about the paper. I believe the paper is above the acceptance threshold for the publication.